# Study of the Phase-Change Thermal-Storage Characteristics of a Solar Collector

**DOI:** 10.3390/ma15217497

**Published:** 2022-10-26

**Authors:** Yuxuan Deng, Jing Xu, Yanna Li, Yanli Zhang, Chunyan Kuang

**Affiliations:** BaiLie School of Petroleum Engineering, Lanzhou City University, Lanzhou 730071, China

**Keywords:** energy, solar energy, concentrator, phase-change heat storage, numerical calculation

## Abstract

A combined solar phase-change thermal-storage heating system is proposed, wherein erythritol is used as the phase-change material (PCM) used to fill the thermal-storage device, and the storage cavity is heated and stored with a disc concentrator. The Solidification/Melting, Volume-of-Fluid (VOF) model of ANSYS Fluent software was used to simulate the phase-change process of erythritol inside the thermal-storage device. The thermal-storage device was designed based on our numerical calculations, and its performance was tested. We found that larger PCM-volume fractions correlated with lower PCM volume-expansion rates and longer total melting times during the heat storage process. When the *φ* value equaled 80%, the PCM solid–liquid-phase interface and temperature distribution were most uniform and showed the best heat storage. In addition, the size of the heat-storage device affected the heat-exchange area, and the total melting time of the PCM decreased and then increased as the width-to-height ratio (I) increased. With this design capacity, the late stage of the charging process of the heat-storage device accounted for 70% of the total time, and the heat energy-utilization rate during the boiling process was 66.3%. Overall, this combined heating system can be considered a very efficient solar energy-utilization terminal for basic domestic energy needs.

## 1. Introduction

Solar energy is an inexhaustible source of renewable energy whose availability far exceeds any conceivable future energy demand and is of strategic importance for China to achieve sustainable development [1,2]. Katsaprakakis used solar energy directly for indoor heating, the proposed solar-combi system can guarantee the 100% annual heating load coverage of the examined building, with an annual contribution from the solar collectors higher than 45% [3,4]. Due to the low density of solar irradiation received on the earth’s surface, solar concentrators are indispensable core devices in high-grade solar energy-utilization systems. For example, they enable smaller area receivers to obtain high-density solar-radiation energy, enhanced energy-utilization efficiency, and reduce construction costs, and are commonly used in industrial and agricultural production and human life [5,6,7,8,9]. Among them, dish solar thermal-power generation systems have the highest solar power-generation efficiency, accounting for approximately 29.4% of the peak solar thermal power generation efficiency worldwide, and have a compact structure, can be easily installed, and show other favorable characteristics, especially in rural or some remote areas, with strong adaptability [10,11,12]. In addition to the use of solar energy for power generation, numerous scholars have studied the direct use of solar energy by converting it into thermal energy for cooking and boiling water through concentrators [13,14,15,16]. Badran et al. [17] designed a portable solar cooker water heater. Aidan [18] constructed and evaluated the performance of a parabolic solar dish cooker in Yola, Nigeria. Panwar et al. [19] tested and analyzed the energy and exergy of a domestic size parabolic solar cooker in actual use. Mohammed [20] presented the design and development of a solar water heater parabolic dish for domestic hot water applications where the water temperature needed to reach 100 °C. Sakhare and Kapatkar [21] presented an experimental platform involving a non-tracking solar paraboloidal dish-concentrating system that could heat water to 215 °C.

Phase-change energy storage involves the use of phase-change materials (PCMs) that store or release energy during the phase-change process to achieve spatiotemporal energy transfer, improve the utilization of energy, and achieve green and efficient energy storage [22,23,24]. Wang et al. [25] found through an experimental study of a paraffin/aluminum foam composite PCM that adding aluminum foam greatly improved the effective thermal conductivity of the composite PCM, accelerated the PCM melting process, and improved the temperature uniformity of the PCM. Zhao et al. [26] conducted an experimental study on the solid–liquid phase-change process and showed that embedding metal bubbles in the PCM can increase the total heat transfer rate by 3–10 times and that the natural convection in the liquid-phase region reduced the temperature difference between the PCM and the wall. Zhang et al. [27] proposed a three-dimensional model consisting of six quadrilateral and eight hexagonal surfaces where the middle A three-dimensional model cut with a ball was simulated to study the melting process of PCMs at different porosities, and a foam metal material with linear variation in the porosity was designed to enhance the heat-transfer process, the results show that the heat storage density increases with the increase of equivalent porosity and metal foams with porosity gradients increase heat storage by enhancing the heat transfer process at the bottom corner. Sundarram et al. [28] used a face-centered-cubic (FCC) structure model and finite volume simulation and found that the foam metal pore size and porosity significantly affected the warming effect of PCM, especially under conditions that promoted high heat production and low convection cooling.

In northwest China, where sunshine hours are relatively abundant, excess heat during the day cannot be stored and heat loss at night is serious, resulting in a mismatch between the heat supply and the spatiotemporal demand. Therefore, the combination of a solar-collector system and a phase-change energy-storage technology is used to achieve the goal of storing energy during the day and discharging heat at night. In this study, the solar energy provided by the collector was equated to a constant heat flow density, and a suitable medium-temperature PCM was selected. The PCM-based transient phase-change process in the thermal-storage system was analyzed using the Solidification/Melting, VOF model of ANSYS Fluent software. The main aims of this work were to utilize numerical calculations to design a thermal-storage device and to analyze the PCM temperature during storage and the release of thermal energy. The results of the study provide a theoretical basis for developing and utilizing medium-temperature PCMs for solar energy.

## 2. Materials and Methods

### 2.1. Physical Model

We used a square cavity energy-storage structure with an outer wall, an inner cavity (150 mm × 150 mm × 170 mm), and a 4 mm thick wall (Figure 1). The inner cavity was filled with PCM and air, and the solar energy was used to heat the bottom of the device so that the PCM underwent a solid–liquid phase change to enable thermal energy storage.

Since the cube structure was symmetrical, the model can be simplified, and the simplified two-dimensional calculation area is shown in Figure 2 with geometric parameters of 150 mm × 170 mm. The bottom was heated by a constant heat flow (*q* = *constant* ≠ 0), and the remaining three walls were all adiabatic (*q* = 0).

### 2.2. Governing Equation

The continuity equation was as follows [29]:(1)$$\frac{\partial \rho }{\partial t}+\nabla \cdot u=0$$
where *ρ* is the density of PCM kg/m^3^, and *u* is the PCM flow rate (m/s).

The momentum equation was as follows [29]:(2)$$\frac{\partial u}{\partial t}+(u\cdot \nabla )(u)=-\frac{1}{\rho }\nabla (P)+\frac{\mu }{\rho }\nabla^{2}u+\frac{1}{\rho }S$$
(3)$$\left\{ \begin{array}{l} S_{x}=\frac{{\left(1-\beta \right)}^{2}}{\left(\beta^{3}+\omega \right)}A_{mush}u\\ S_{y}=\frac{{\left(1-\beta \right)}^{2}}{\left(\beta^{3}+\omega \right)}A_{mush}u-\rho g\alpha (T-T_{ref}) \end{array}\right.$$
where *β* is the liquid-phase rate in the range of 0–1, *μ* is the dynamic viscosity (Pa·S), *A_mush_* is the paste-phase region constant (generally taken as 10^5^), *ω* is a constant used to avoid introducing a value of 0 into the formula (generally taken as 10^4^–10^7^), and α is the coefficient of thermal expansion.

The energy equation was as follows [30]:(4)$$\frac{\partial }{\partial x}(\rho H)+\nabla (\rho uH)=\nabla \cdot (k\nabla T)+S$$
(5)$$\left\{ \begin{array}{l} H=h+∆H\\ h=h_{ref}+\int_{T_{ref}}^{T}c_{p}dT\\ ∆H=\beta L \end{array}\right.$$
where *H* is the total enthalpy (kJ/kg) of the PCM, *h_ref_* is the reference enthalpy (kJ/kg), *h* is the sensible enthalpy (kJ/kg), *L* is the latent heat value (kJ/kg), *k* is the thermal conductivity (W/m·K), *S* is the relevant source term, and *cp* is the constant pressure-specific heat capacity of the PCM, expressed as J/(kg·K).

### 2.3. Materials

The type of PCM and the heat-storage properties determine the heat-storage density and heat-transfer characteristics of the thermal-storage system and greatly impact the heat-storage efficiency of a storage device [31]. Table 1 shows several commonly used medium- and low-temperature PCMs and their physical parameters [32,33,34]. Inspection of Table 1 reveals that the latent heat of phase change of erythritol is 339 kJ/kg. Thus, selecting erythritol for phase-change heat storage can reduce the mass of the heat-storage material and the volume of the heat-storage device while maintaining the same heat-flow density, reducing the transportation cost per unit of heat storage, and increasing the economy of the solar energy-utilization system. Therefore, erythritol was analyzed in this study.

### 2.4. Numerical Simulation

To facilitate the calculations, the following assumptions were made: (1) The exterior of the thermal storage device was wrapped with insulation material, enabling heat dissipation from the exterior wall to the environment to be ignored, (2) the collector heated the bottom of the thermal storage device, so the boundary conditions were considered to have a constant heat-flow density, (3) the PCM was isotropic, assuming that the air within behaved as an ideal gas, (4) the melted erythritol was an incompressible Newtonian fluid and the effect of the buoyancy force generated by the density adhered to the Boussinesq model.

#### 2.4.1. Grid Independence

Structured mesh dissection was performed using ICEM CFD software for the two-dimensional computational domain shown in Figure 2. To reduce the influence of the grid size on the calculated results, grid-independent analysis was conducted to compare the melting of erythritol under four grid schemes, namely 14,256, 36,036, 53,976, and 69,936 (Figure 3). The maximum deviations of the erythritol liquid-phase volume fraction were 5.72% and 6.72% when the grid number was increased from 36,036 to 53,976 or 69,936, respectively, and the influence of the grid size on the calculated results is small. The 36,036 grid scheme was finally selected for the numerical calculations by combining the calculation accuracy and resources.

#### 2.4.2. Boundary Conditions and Setting

The Solidification/Melting, VOF model of ANSYS Fluent software was used to simulate the phase-change process of erythritol in the thermal-storage device. A two-dimensional non-stationary, implicit solver was used, the second order upwind format was chosen to discretize the momentum and energy equations, and the PRESTO format was used to discretize the pressure gradient. The SIMPLE algorithm was used for pressure and velocity coupling, the partial-relaxation factor was adjusted downward to 0.5 to ensure better convergence, and the time step was set to 0.01 s for the calculations. The second type of boundary condition, i.e., a constant heat-flow density (*q* = 20,000 W/m^2^) was used as the heat source to heat the bottom wall, and the other walls were set to an adiabatic boundary condition, i.e., *q* = 0. The temperatures of all parts of the thermal-storage device were equal at the initial moment, i.e., the initial temperature was T_0_ = 298 K. The PCMs were set according to the physical parameters of erythritol presented in Table 1.

### 2.5. Experiment Test

The design of the thermal-storage device was based on the results of the numerical calculations, the size of the square cavity was 160 × 160 × 160 mm, the solid phase PCM filled 80% of the volume of the square cavity, and the square cavity was placed in the insulation box filled with a thermal insulation layer (Figure 4). In the heat-storage phase, the solar collector was used to heat the thermal-storage device, and the PCM was heated to melt and absorb heat to achieve heat storage in the thermal-storage device. During the exothermic phase, the thermal storage device released a large amount of heat to heat the water in the heat exchanger (kettle) on the user side to drive the release of heat on the user side. Its performance was tested from the following two aspects: (1) The temperature change of PCM during heat storage and (2) the temperature change of the PCM and heated medium (4.5 L of water) during the exothermic process.

## 3. Results and Discussion

### 3.1. Model Validation

To verify the reliability of the calculated results, the temperatures at two characteristic points A (0.5, 0.25) and B (0.5, 0.75) on the centerline of the device were monitored and compared with the results of the numerical calculations (Figure 5). It can be seen that the simulated values at the beginning of melting matched well with the experimental values. After heating for a certain time, the temperature of each PCM layer reached the melting point (from the bottom to the top) and completed the phase change layer by layer. When the PCM was completely melted, the temperature of each layer tended to be consistent and rise at the same time. During the middle and late stages of heating, the natural convection and overheating phenomena generated during the experimental process made the simulated and experimental values inaccurate. Overall, the changes tended to be basically the same. At the end of the melting, the errors at the two points were 1.6% and 1.8%, respectively, which were within the acceptable range, so the results of our results can be considered reliable.

### 3.2. Effect of the PCM Volume Fraction on the Thermal-Storage Process

The PCM density decreased during the solid–liquid phase change. Therefore, volume expansion was inevitable, and the PCM volume fraction (*φ*) had a large effect on the expansion. In this study, the phase field and thermal field were analyzed at different *φ* values (40%, 50%, 60%, 70%, 80%, and 90%). The formula for *φ* was as follows:(6)$$\varphi =\frac{V_{\mathrm{pcm}}}{V}\times 100\%$$
where *V*_pcm_ and *V* denote the solid-phase volume and the total volume of the PCM cavity, respectively.

Figure 6 shows the phase-interface variation for different *φ* values with liquid-phase rates of *β* = 0.5 and *β* = 1 (half melted and fully melted). The red shading indicates the liquid phase, the blue shading indicates the solid phase, the gray shading indicates the gas phase, the yellow shading represents paste-like PCM between the solid and liquid phases, and the white dashed line indicates the initial position of the solid PCM. At *β* = 0.5, the gas–solid intersection overlapped with the white dashed line, at which time the PCM had almost no volume expansion. At *β* = 0.5, with increasing *φ* values, the solid–liquid interface fluctuation was enhanced, which was due to the natural convection being enhanced by buoyancy and gravity as the liquid PCM increased. At *φ* = 80%, the fluctuation of the solid–liquid phase interface was maximal, and the heat transfer in the device was dominated by natural convection. At *φ* = 90%, a two-phase pasty region appeared near the middle axis. The main reasons that the PCM acquired a pasty appearance are as follows. When the device was filled with 90% PCM and 10% air, an excess amount of solid PCM was influenced by gravity, and as melting proceeded, the area of liquid PCM in the device increased, and the natural convection phenomenon was enhanced. Thus, under the action of gravity and buoyancy, the liquid PCM moved along the sides of the solid PCM to the top of the device, whereas the solid PCM located above moved from the middle to the bottom of the device. Consequently, the liquid PCM in the middle region of the device contacted the solid PCM at a lower temperature during the flow, becoming cooled again and changing from a liquid phase to a paste, or even to a solid phase, which led to an unstable and uneven heat transfer. At *β* = 1.0, the PCM had melted completely, and the gas–solid intersection evolved into a gas–liquid intersection with a waveform. The gas–liquid-intersection interface was higher than the white dashed line at different *φ* values, and it approached the white dashed line as the *φ* value increased. These results indicate that the PCM expanded to different degrees during the process of 0.5 < *β* ≤ 1, and the amount of expansion was lower with increasing *φ* values. This outcome was due to the fact that the corresponding air volume decreased as the *φ* value increased with the constant total volume of the device, resulting in a restricted space for expansion of the liquid PCM volume.

Figure 7 shows the temperature variation of the liquid phase at rates of *β* = 0.5 and *β* = 1 for different *φ* values. At different *φ* values, the PCM had the highest temperature near the bottom heating wall, followed by the side wall region, because the thermal conductivity of the metal was larger than that of the PCM, and the PCM temperature near the device wall was first heated in order to transfer heat to the low-temperature region. The PCM with a higher temperature near the wall was “sprayed” onto the lower-temperature region inside, and as *φ* increased, the intensity and area of “spraying” became larger, and the temperature distribution became more uneven. At *β* = 0.5, a local columnar low-temperature region appeared in the middle position, and as *φ* increased, the temperature gradient in this region gradually decreased, and the heating became more uniform. However, when *φ* continued to increase to 90%, a localized low-temperature region reappeared in the middle region, resulting from the natural convection generated by the liquid PCM, which caused the PCM with a higher density and a lower temperature to move upward from the sides, whereas the liquid PCM with a lower temperature and a higher density moved downward from the middle, consistent with the phase-interface distribution at *β* = 0.5 shown in Figure 5. As the temperature gradually decreased, the temperature gradient was uniformly distributed, and the gas–liquid isothermal surface was consistent with the gas–liquid phase interface at *β* = 1.0 (Figure 5), although a columnar high-temperature zone appeared in the middle axis. With increasing *φ* values, irregular high-temperature zones were generated at the bottom wall surface with an uneven temperature distribution, which was due to continuous heat input during the energy-storage process, resulting in the liquid PCM that melted first near the heat-transfer surface and stored heat through absorption as the temperature increased. When the *φ* was increased to 90%, the temperature distribution produced a small vortex, perhaps because the phase interface was too close to the top of the device, resulting in a smaller space for the air layer above and, as the air squeezed the PCM fluid more unevenly, it produced a small squeezed area with a small vortex, resulting in an irregular temperature-distribution pattern.

Figure 8 shows the relationship between the *φ* value and the melting time of PCM. At the early stage of melting (t < 1000 s), the *φ* had little effect on the change in the volume of liquid PCM because less liquid PCM was present during that period, the convection intensity was small, and heat transfer was dominated by heat conduction. When the time increases to 1000 s, the slope of the *φ* = 40% working-condition curve was the first to increase, reaching the peak melting period, and the natural convection intensity increased. As the melting continued, the *φ* = 50%−90% working conditions also reached peak melting periods, although 1.82-, 2.22-, 2.46-, 2.37-, and 3.20-fold more time was required to reach the melting peak, respectively, compared to when *φ* equaled 40%. After the peak period (except for the *φ* = 80% condition), the liquid PCM volume-fraction change curve for each condition showed an inflection point, and the melting reached the weakening stage, whereupon the PCM melting rate decreased sharply and then rebounded, and slowly decreased with over time until the liquid PCM volume fraction no longer increased and reached stability and the latent heat storage ended. The peak-like change in the melting rate during the decay phase occurred because the liquid PCM contacted the lower-temperature air above the device and the liquid PCM solidified when it cooled, resulting in a sudden decrease in the liquid PCM volume fraction, but as the melting continued, the solid PCM continued to melt until melting was complete. With increasing *φ* values, the time required to completely melt the PCM became longer, but the time required for the *φ* = 80% condition was shorter than that for the 60% condition by 256.5 s. These results are consistent with the results of the previous analysis of phase and thermal fields, which indicated that an energy-storage device set at *φ* = 80% had better heat transfer and the best overall thermal-storage performance during the heat-storage process.

### 3.3. Effect of the Square Cavity Structure on the Heat-Storage Process

The results presented in Section 3.2 show that the flow of the PCM was more stable and that the heat-transfer efficiency was higher when *φ* equaled 80%. Therefore, we used *φ* = 80% for the solid-phase erythritol to fill the accumulation square cavity and investigate the effect of square cavity structure on the heat-storage process. With the volume of the computational domain kept constant, three different width-to-height ratios were studied (Table 2). The width-to-height was expressed as the dimensionless number *I*.
(7)$$I=\frac{W}{H}$$

Figure 9 shows the PCM phase-interface distributions of three energy-storage devices with different aspect ratios at four different points during the melting process. The longer the melting time, the more liquid PCM was present in the device. At any moment, the liquid PCM area first increased and then decreased when the I increased. When I equaled 0.88, the liquid PCM was high in the middle and low on both sides, and a paste-like area appeared between the solid–liquid interface when the melting reached the later stage. When I equaled 1.0, the liquid PCM area was larger than the other two working conditions at any moment, and the interface distribution was the most uniform. When *I* equaled 1.23, the phase-interface distribution was low in the middle and high on both sides, and a two-phase paste-like region appeared in the middle region. The reason for this phenomenon is as follows. More PCM in the horizontal direction made the PCM in the middle area of the device move away from the heating wall, and side wall surface, the heat-exchange area near the bottom heating element increased, and the heat-transfer became faster, which increased the liquid phase of PCM in the device. Thus, natural convection was enhanced, and due to gravity and buoyancy, the liquid PCM moved along the left and right sides toward the top of the device, causing the PCM to solidify (due to the lower temperature above the device) and move from the middle toward the bottom of the device. By the late stage of melting, all phase-interface distribution tended to be horizontally distributed.

Figure 10 shows the PCM temperature distributions in the three energy-storage devices with different aspect ratios at four different points during the melting process. The temperature distribution is consistent overall with the phase-interface distributions shown in Figure 8. When *I* equaled 0.88, several intermittent bumps with higher temperatures appeared on the bottom heating wall, and a local low-temperature zone was generated near the center line. As the melting process proceeded, the high-temperature region at the bottom became continuous, and the localized low-temperature region at the centerline was replaced with a high-temperature “jet” of liquid PCM. When *I* equaled 1.0, a “tree-like” high-temperature zone appeared during the early stage of melting, resulting in a large temperature gradient, which favored the melting process. The temperature distribution was more uniform during the middle of the melting process, similar to when *I* equaled 0.88. However, when melting reached the later stage, the distributions of the high- and low-temperature zones became more regular, which favored heat transfer. When *I* equaled 1.23, a columnar low-temperature region appeared in the middle region of the energy-storage device during the early melting stage. As melting proceeded, the middle low-temperature region gradually increased in size and expanded to both sides, although the temperature in the liquid-phase region tended to stabilize during the late melting stage, and a large isothermal region appeared.

Figure 11 shows the relationship between the *I* value and PCM melting time. The melting process could be roughly divided into three stages. t < 300 s was the pre-melting period (blue dashed box in the figure) when the curves under the three operating conditions changed linearly with a small slope. When t was >300 s, the slope of the curve increased until 2766 s when the slope basically remained the same, and the period was designated as the middle melting period. The reason for this phenomenon is that before t = 300 s, the PCM heat-transfer mode was dominated by conduction, and the melting was slow. However, when entering the middle melting period, the heat-transfer mode of the PCM displayed both conduction and natural convection, with natural convection dominating, such that the slope of the curve and the melting rate increased. When t was >2766 s, the melting reached the late stage, when the natural convection effect between the liquid PCM disappeared, the melting rate decreased rapidly, and the slope of the curve suddenly decreased. In the case where *I* equaled 1.23, a wave change occurred when the melting proceeded to t = 2000 s. The reason for this phenomenon may be as follows. During the melting process, the low-temperature air above the device contacted the high-temperature liquid PCM, which caused the liquid PCM to solidify. As the melting proceeded, the temperature of the solidified PCM gradually increased and melted back to the liquid state again, and the time required for complete melting was t = 3000 s. Overall, our results suggest that when *I* equaled 1.0, the melting rate changed steadily, and the time required for complete melting was the shortest, indicating the best heat-transfer effect occurred under that condition.

### 3.4. Experimental Results

Figure 12 shows temperature-change curves for the PCM during the heat-storage process, where curve A represents the temperature-change law of the PCM from the initial ambient temperature (6.6 °C) to the end of the heat storage, and curve B represents the temperature-change law of the PCM when the heat-storage device continued to be heated after a period of use. The graph shows that 10 h was required to fully charge the thermal-storage device to the design capacity and that the PCM reached a maximum temperature of 523 °C after the charging ended. After a period of use, the temperature of the PCM dropped to 323 °C. A period of approximately 7 h was required before charging could be continued, and the PCM reached a maximum temperature of 563 °C after charging. Because the test was conducted in an open external environment, the heat-flow density of each heating step was not guaranteed to be equal during different time periods, and the error between the maximum temperature that the PCM could reach in both tests was ignored. The slope of curve B is significantly smaller than that of curve A and appeared flatter, indicating that the PCM had a slower rate during the process of heating from 323 °C to 563 °C and taking 70% of the time for charging. Figure 13 represents the temperature-change curve observed for the PCM and water during the exothermic process when the thermal-storage device was used to heat a kettle. Twenty minutes was required to heat 4.5 L of water from 20 °C to 88 °C, and the PCM temperature decreased from 575 °C to 440 °C. The physical parameters of the PCM and water were calculated as 66.3% of the energy required to warm up the water when PCM released heat energy. In the open environment outside, the heat-storage device showed more heat leakage, which lowered the performance of the heat-storage device.

## 4. Conclusions

The PCM volume fraction *φ* greatly influenced the heat-flow and heat-transfer characteristics of the PCM during the phase-change process. The larger the *φ* value was, the lower the PCM volume-expansion rate during the heat-storage process and the longer the total melting time. At *φ* = 80%, the best heat-storage effect was achieved due to the rapid development of Rayleigh convection, which rendered the PCM solid–liquid phase interface and temperature distribution more uniform inside the device. The aspect ratio (*I*) also greatly influenced the heat-flow and heat-transfer characteristics of the PCM during the phase-change process. When *I* equaled 1.0, the volume fraction of liquid PCM was larger than that found under the other two working conditions at any moment, the interface distribution was the most uniform, the melting rate changed steadily, and the time required for complete melting was the shortest, indicating the best heat transfer occurred under that condition. We also found that, with the current design capacity, 10 h was required to fully charge the heat-storage device, the middle and late stages took 70% of the overall charging time, the heat leakage of the device was relatively low with the current design, and the heat-utilization rate during the boiling process was only 66.3%. Overall, in a typical rural living environment, this type of thermal-storage system can be considered a very efficient solar energy-utilization terminal, where the morning cooking schedule can be completed by 10 a.m., with off-peak hours starting at 11 a.m., during which the stove temperature gradually rises. It usually takes about 7 h to reach a temperature suitable for cooking again. Rural dinners usually begin at 6 p.m., which is the time when the stove reaches its maximum temperature.

## Figures and Tables

**Figure 1 materials-15-07497-f001:**
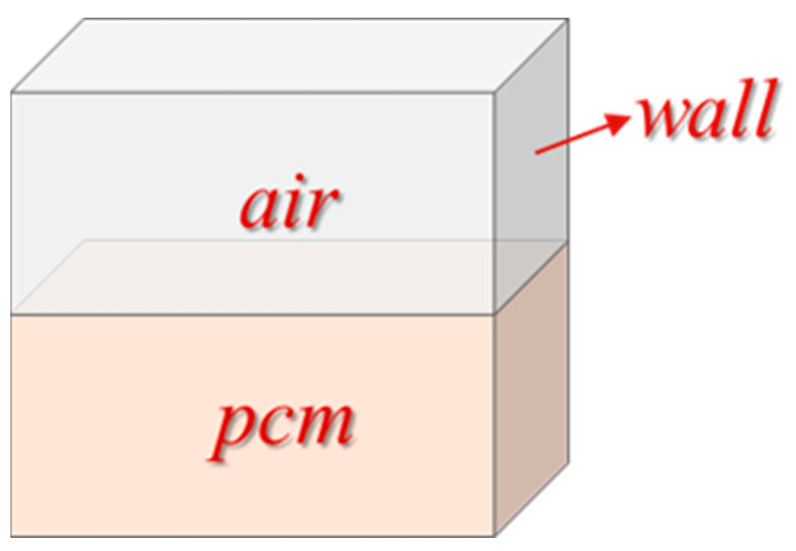
Physical model.

**Figure 2 materials-15-07497-f002:**
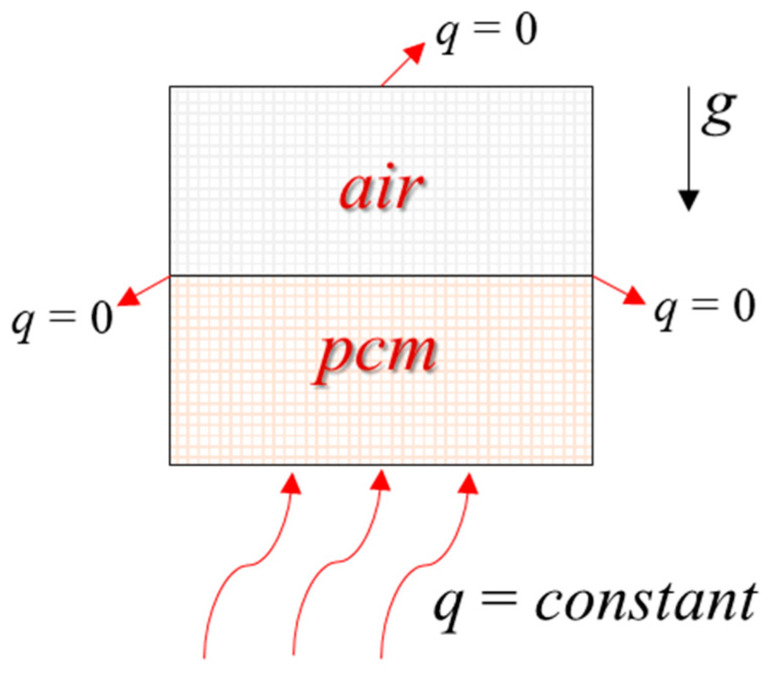
Computational model.

**Figure 3 materials-15-07497-f003:**
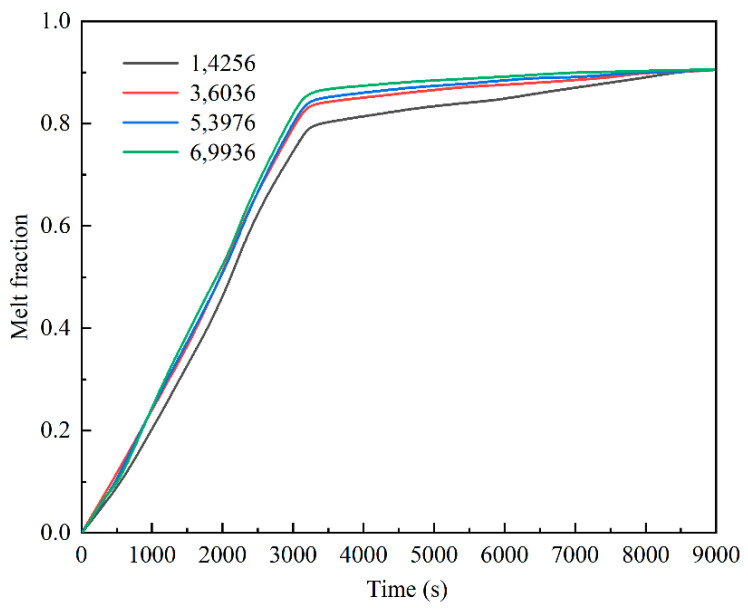
Grid-independence verification.

**Figure 4 materials-15-07497-f004:**
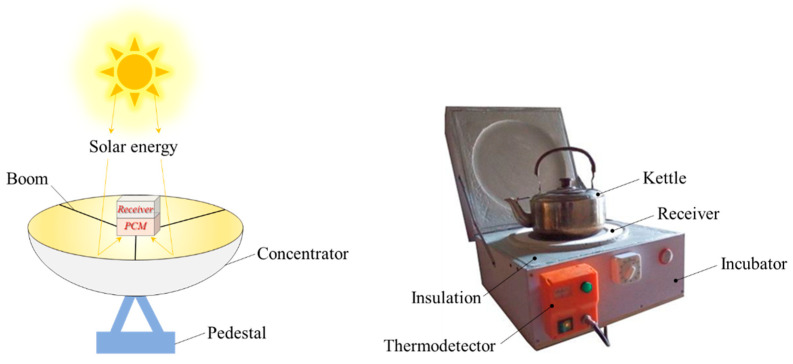
Experimental test.

**Figure 5 materials-15-07497-f005:**
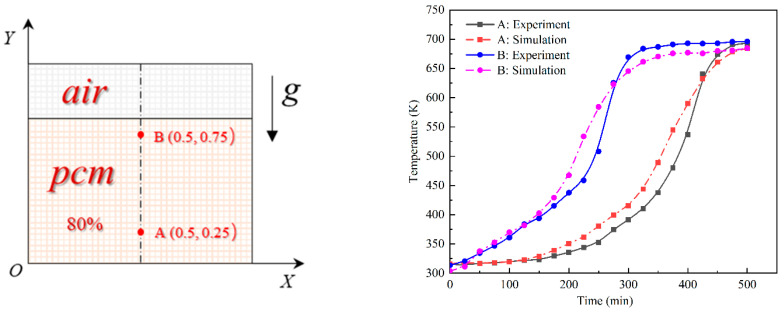
Comparison between the experimental and simulated results.

**Figure 6 materials-15-07497-f006:**
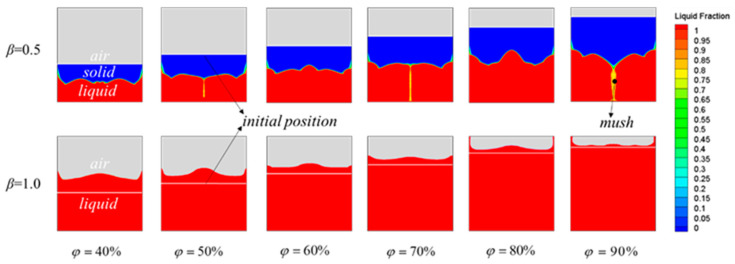
Phase fields at different *φ* values.

**Figure 7 materials-15-07497-f007:**
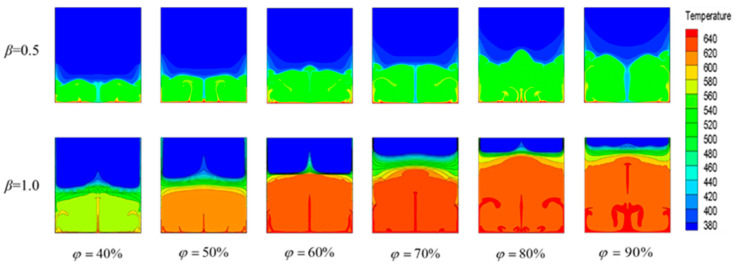
Temperature fields at different *φ* values.

**Figure 8 materials-15-07497-f008:**
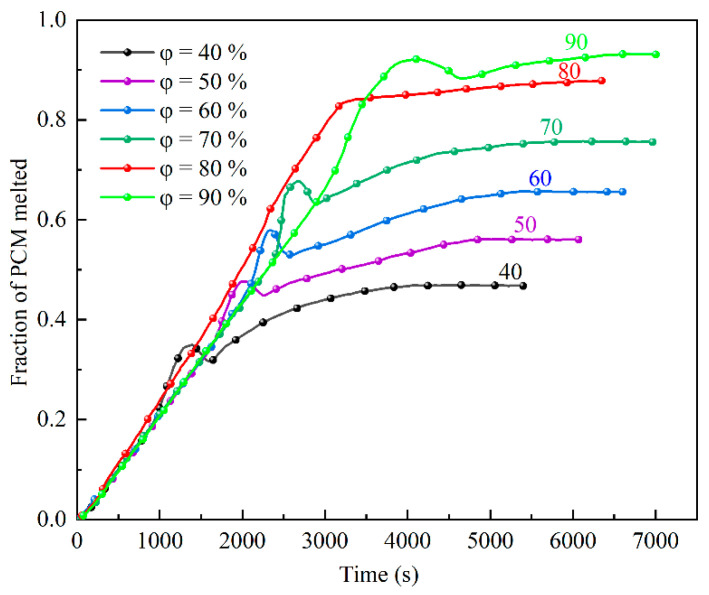
PCM melting time at different *φ* values.

**Figure 9 materials-15-07497-f009:**
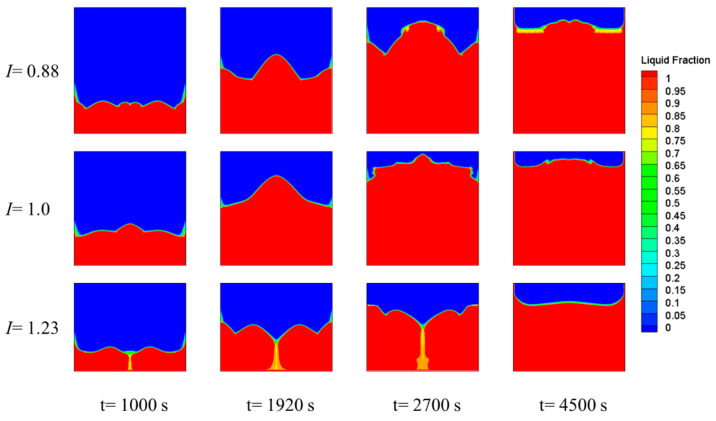
Phase fields for the melting process at different *I* values.

**Figure 10 materials-15-07497-f010:**
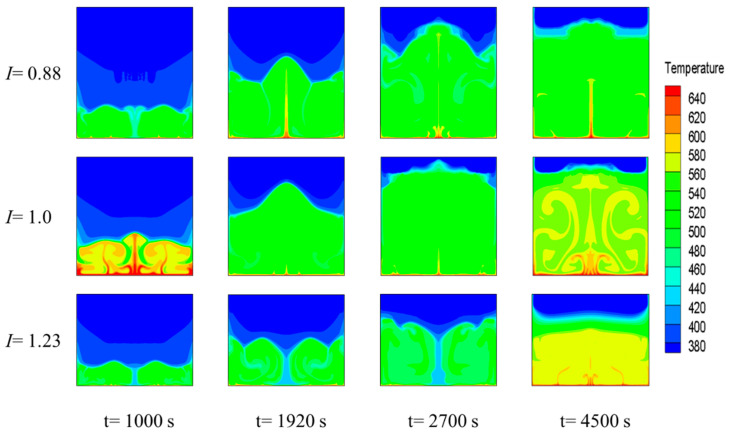
Temperature fields for the melting process under different *I* values.

**Figure 11 materials-15-07497-f011:**
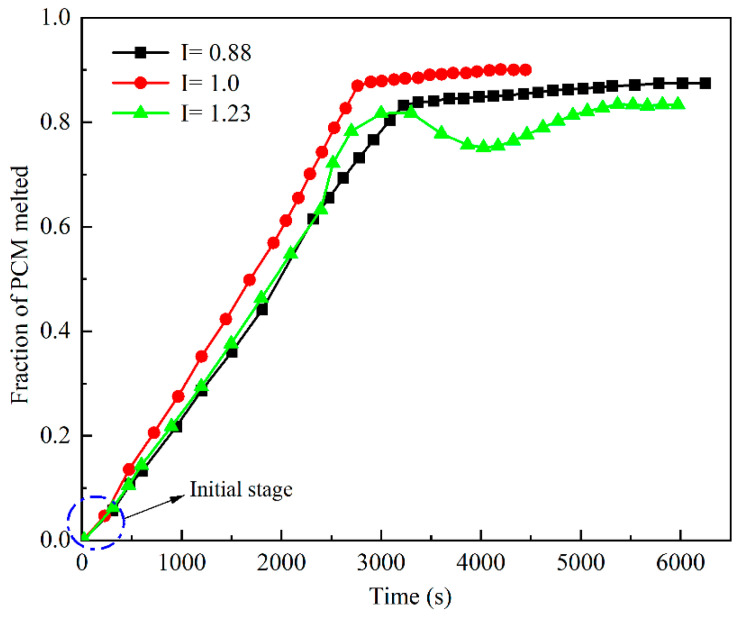
Melting times of PCM at different *I* values.

**Figure 12 materials-15-07497-f012:**
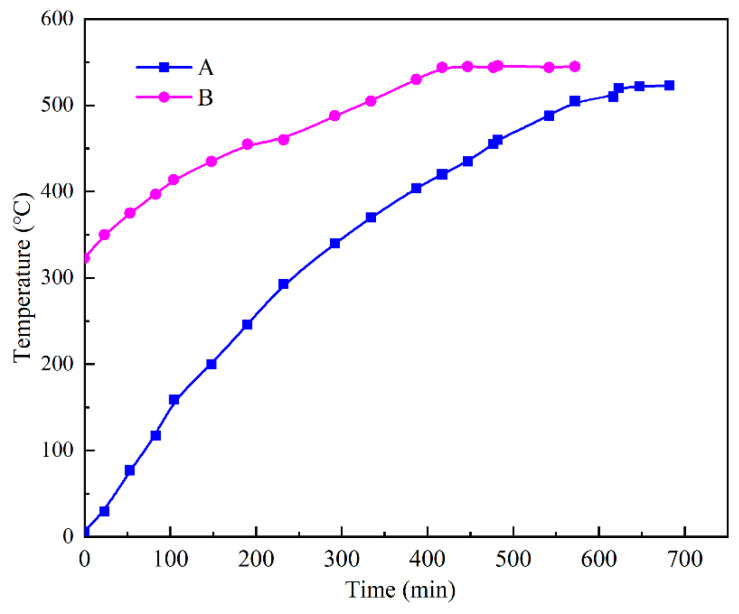
Temperature change of PCM during heat storage.

**Figure 13 materials-15-07497-f013:**
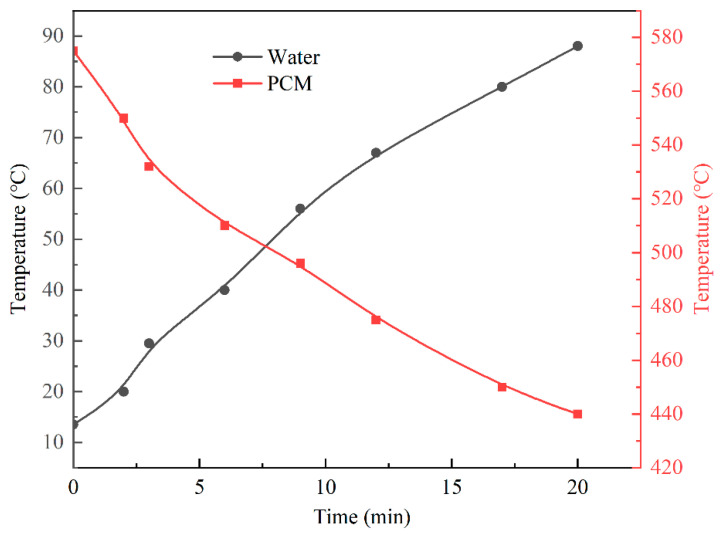
Exothermic process for the PCM when the water temperature changed.

**Table 1 materials-15-07497-t001:** Physical parameters.

Number	Material	Phase-Change Temperature (°C)	Latent Heat of Phase Change (kJ/kg)	Thermal Conductivity (W/m·K)	Density (kg/m^3^)
1	Fatty acid	70.0	186.5	0.172	848
2	Barium hydroxide octahydrate	78.0	265.7	0.653	1660
3	Naphthalene	80.0	147.7	0.132	976
4	Magnesium chloride hexahydrate	117	168.0	0.570	1450
5	Erythritol	118	339.0	0.326	1330

**Table 2 materials-15-07497-t002:** Aspect ratio of the device.

Working Condition	*W*/mm	*H*/mm	*I*
1	150.0	170.0	0.88
2	156.4	156.4	1.00
3	180.0	145.8	1.23

## Data Availability

All the data is already in the article.

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
