# Peer review of "Study of the Phase-Change Thermal-Storage Characteristics of a Solar Collector"

_materials, 2022, doi:10.3390/ma15217497_

Round 1
Reviewer 1 Report
In the review of the article titled: Study of the phase-change thermal-storage characteristics of a solar collector, authors have presented the work with significant detail. They have addressed the results very well in good descriptive way. I would like to see this article publish but after some minor modifications as follow;
1. In Figure 3, Grid -independence verification, Can you please calculate its melt fraction for longer hour’s e.g. 24 hours; 36 hours etc and provide comparative studies.
2. To verify the reliability of the calculated results, only temperatures at two characteristic points A (0.5, 0.25) and B (0.5, 0.75) on the centerline of the device were monitored and compared with the results of the numerical calculations. Can you please provide other characteristics results e.g volume and pressure etc?
3. Authors have shown Phase fields at different ϕ values, what about 120, 150 or 180 angles observation? Can you please perform phase field at these angles.
4. How do you calculate the expect ratio of the device numerically?
5. Fraction of PCM melted at 90 degree looks so weird? Can you add more explanation for this unusual behavior at this phase?
6. If phase interface becomes too close to the top of the device, how it will affect the temperature distribution pattern?
7. At 120 to 180 ϕ values, how PCM temperature will affect the bottom heating wall, as well as side wall region?
8. How can we fully charge the thermal storage device to the design capacity?

Author Response
Dear reviewer,
Thank you very much for your constructive comments on our paper. We have given detailed answers to your comments. Please refer to the attachment.
Sincerely,
Yuxuan Deng
BaiLie School of Petroleum Engineering, Lanzhou City University, Lanzhou City, 730000, China
Telephone:+8615908150884
dengyuxuan@lzcu.edu.cn

Reviewer 2 Report
The manuscript entitled “Study of the phase-change thermal-storage characteristics of a
solar collector” has been submitted by authors. Some issues to be addressed which will improve the quality of manuscript. Therefore, I recommend this work could be published after the major revision
1. Author should clearly write down novelty of this paper
2. Check the format of the reference and correct all the errors.
3. All the references mentioned in the paper should be cited in the text or vice-versa.
4. The table and figures heading should be incorporated and discussed in the text.
5. The English composition requires many improvements. The authors should proofread the manuscript carefully to minimize grammatical errors.
6. Please enhance revolution and make all figure text readable.
7. The conclusion is too long; please make it short and to the point.

Author Response

(The authors gave the same response as above.)

Reviewer 3 Report
The authors report on the phase change performance of erythritol for solar concentrating power storage based on simulation and experimental data. The manuscript is in general well written, but a few points requiring attention as highlighted in the attached pdf file. There are valuable simulations presented in the manuscript, however, the introduction should be strengthened to provide the rationale for selecting the specific solar concentrator and phase change material over the numerous alternatives. After this is introduced, the presented results should be compared with the alternatives in terms of performance and discussed, to provide suggestions for further development of the phase change materials and designs.

Author Response

(The authors gave the same response as above.)

Round 2
Reviewer 2 Report
The author solves all comments very carefully. now it's ready to accept in present form.
Reviewer 3 Report
The authors successfully addressed the concerns of the review. Before the paper is published, please consider adding replies to the concerns in the text of the manuscript. Especially for responses R1-14 and R1-15 with reference to the relevant published data, to aid towards a better understanding by the reader.